# Somatic point mutations are enriched in non-coding RNAs with possible regulatory function in breast cancer

Narges Rezaie [1,10], Masroor Bayati[2,10], Mehrab Hamidi[2], Maedeh Sadat Tahaei[2], Sadegh Khorasani[2], Nigel H. Lovell[3], James Breen[4,5,6], Hamid R. Rabiee [2✉] & Hamid Alinejad-Rokny [7,8,9✉]

Non-coding RNAs (ncRNAs) form a large portion of the mammalian genome. However, their biological functions are poorly characterized in cancers. In this study, using a newly developed tool, SomaGene, we analyze de novo somatic point mutations from the International Cancer Genome Consortium (ICGC) whole-genome sequencing data of 1,855 breast cancer samples. We identify 1030 candidates of ncRNAs that are significantly and explicitly mutated in breast cancer samples. By integrating data from the ENCODE regulatory features and FANTOM5 expression atlas, we show that the candidate ncRNAs significantly enrich active chromatin histone marks (1.9 times), CTCF binding sites (2.45 times), DNase accessibility (1.76 times), HMM predicted enhancers (2.26 times) and eQTL polymorphisms (1.77 times). Importantly, we show that the 1030 ncRNAs contain a much higher level (3.64 times) of breast cancer-associated genome-wide association (GWAS) single nucleotide polymorphisms (SNPs) than genome-wide expectation. Such enrichment has not been seen with GWAS SNPs from other cancers. Using breast cell line related Hi-C data, we then show that 82% of our candidate ncRNAs (1.9 times) significantly interact with the promoter of protein-coding genes, including previously known cancer-associated genes, suggesting the critical role of candidate ncRNA genes in the activation of essential regulators of development and differentiation in breast cancer. We provide an extensive web-based resource (https://www.ihealthe.unsw.edu.au/research) to communicate our results with the research community. Our list of breast cancer-specific ncRNA genes has the potential to provide a better understanding of the underlying genetic causes of breast cancer. Lastly, the tool developed in this study can be used to analyze somatic mutations in all cancers.

[1] Center for Complex Biological Systems, University of California Irvine, Irvine, CA 92697, USA. [2] Bioinformatics and Computational Biology Lab, Department of Computer Engineering, Sharif University of Technology, Tehran 11365, Iran. [3] Tyree Institute of Health Engineering and The Graduate School of Biomedical Engineering, UNSW Sydney, Sydney, NSW 2052, Australia. [4] South Australian Health & Medical Research Institute, Adelaide, SA 5000, Australia. [5] Robinson Research Institute, University of Adelaide, Adelaide, SA 5006, Australia. [6] Bioinformatics Hub, University of Adelaide, Adelaide, SA 5006, Australia. [7] BioMedical Machine Learning Lab (BML), The Graduate School of Biomedical Engineering, UNSW Sydney, Sydney, NSW 2052, Australia. [8] UNSW Data Science Hub, The University of New South Wales (UNSW Sydney), Sydney, NSW 2052, Australia. [9] Health Data Analytics Program, AI-enabled Processes (AIP) Research Centre, Macquarie University, Sydney, NSW 2109, Australia. [10] These authors contributed equally: Narges Rezaie, Masroor Bayati. ✉email: rabiee@sharif.edu; h.alinejad@unsw.edu.au

Breast cancer is the most common cancer in women that has the highest frequently leading cause of cancer-related mortality amongst females worldwide[1]. A deeper understanding of the underlying mechanisms of breast cancer genetics and pathogenesis can be used to detect early-stage cancer to reduce morbidity and mortality due to breast cancer[2]. Over the last 10 years, high-throughput sequencing has comprehensively investigated the underlying genetic mechanisms that initiate or drive cancer progression and revealed many cancer-associated mutations. The International Cancer Genome Consortium (ICGC)[3] has provided an extensive catalog of somatic mutations for various cancer types. This information has enabled researchers to characterize numerous protein-coding genes essential in cancer progression[3–7].

Although most studies have mainly focused on protein-coding genes in investigating driver mutations in cancer, several lines of evidence imply that about 80% of the genome is biochemically functional[8], indicating there are many mutations in non-coding regions that need to be investigated. A large portion of non-coding regions operates as regulators of oncogenes[9]. In addition, around 93% of disease-associated genome-wide association single nucleotide polymorphisms (GWAS SNPs) are located within these regions[10,11], which can significantly influence gene expression of coding and non-coding genes.

Long non-coding RNAs (lncRNAs), an influential class of non-coding transcripts with more than 200 nucleotides, are potential cancer progression indicators and are emerging as diagnostic biomarkers in cancer and other diseases[12–17]. For example, GAS5 is one of the lncRNAs considerably downregulated in breast cancer[18]. HOTAIR is another well-known lncRNA upregulated in breast cancer, contributing to aberrant histone H3K27 methylation and cancer metastasis[19,20]. Furthermore, lncRNAs contribute to various regulatory activities in the cell, such as regulating gene expression via interaction with other chromatin regulatory proteins[21], functioning as active enhancers[22,23], and regulating chromatin structure[17,24,25]. Despite various studies on the impact of non-coding somatic mutations occurring in ncRNAs, the role of such ncRNAs has remained underexplored in breast cancer.

In this study, we developed a new tool, SomaGene, to identify 1030 ncRNAs that are significantly and specifically mutated in breast cancer. Using SomaGene, we show that candidate ncRNAs identified in our study are enriched considerably for regulatory features (e.g., breast-specific H3K27ac, H3K4me1, CTCF, DNase hypersensitive sites (DHS), and enhancer marks). Notably, we show that our breast cancer candidate ncRNAs have a much higher fraction of GWAS SNPs and expression of quantitative trait loci (eQTL) polymorphisms. Finally, our analyses on high-throughput chromosome conformation capture (Hi-C) data from the Human Mammary Epithelial Cell (HMEC) indicate that many of our candidate ncRNAs significantly interact with at least one protein-coding gene, which may suggest a potential enhancer role for these ncRNAs. An overview of the pipeline used in this study is shown in Fig. 1.

We also compared enrichment of regulatory features, GWAS SNPs, and eQTL polymorphisms in the candidate set of ncRNAs with the 2nd, 3rd and last sets of ncRNAs (each set contains 1030 ncRNAs). We first sorted ncRNAs based on their mutational P value to identify 2nd, 3rd, and last sets of ncRNAs. i.e., ncRNAs that significantly mutated in breast cancer samples are positioned in the top places of the list (referring to the 1030 ncRNAs as the candidate set). We then defined the next 1030 ncRNAs after the candidate ones, in the sorted list, as the 2nd set, and the subsequent 1030 ncRNAs after the 2nd set, as the 3rd set of ncRNAs. We also defined the last 1030 ncRNAs in the sorted list as the last set of ncRNAs (i.e., those ncRNAs with the worse mutational P value in breast cancer).

We provide a list of non-coding genes with their mutational enrichment P value and annotated genomic signals at http://www.ncrna.ictic.sharif.edu and https://www.ihealthe.unsw.edu.au/research. SomaGene as an open-source R package is also available at https://github.com/bcb-sut/SomaGene.

## Results

To have a comprehensive list of ncRNAs, we used a combined list of ncRNAs provided by the FANTOMCAT[26], Ensembl[27] consortia, and an atlas of ncRNAs[28] (see method section). This includes ncRNAs from different types inclusive of pseudogene (22.9%), lncRNA intergenic (21.4%), long intergenic ncRNAs (5.6%), lncRNA divergent (13.4%), antisense (3.3%), lncRNA sense intronic (6%), miRNA (5%), misc RNA (3%), lncRNA antisense (4.8%). A full list of ncRNAs is provided in Fig. 2a. Somatic mutations from 17 cancer types were downloaded from ICGC[29], including 1855 breast cancer samples containing 17,163,482 single point somatic mutations and 10,460 samples with other cancers containing 67,752,271 somatic point mutations inside the ncRNA regions.

**Background model to identify significant non-coding RNAs in breast cancer.** To identify the significantly mutated ncRNAs in breast cancer samples, we counted the number of samples with somatic mutations in ncRNA from 1855 breast cancer samples and 10,460 samples with other cancers. We then calculated a P value for each ncRNA using Fisher's exact test (see method section). We calculated P values for 1,000,000 random permutations of breast/non-breast labels for each ncRNA to identify significantly mutated ncRNAs. We estimated that an association's probability emerges by chance (see method section). This provides a threshold for each ncRNA separately (Fig. 3). As a result, we identified 1030 ncRNAs (99% confidence interval—see method section) that significantly mutated in breast cancer samples (Supplementary Data 1). Looking into our candidate ncRNAs revealed that 27.2% of them are lncRNAs intergenic, 18.1% pseudogene, 3.8% long intergenic non-coding RNAs (lincRNA), 20.8% lncRNA divergent, 2.5% antisense, and 6.6% lncRNA sense intronic (Fig. 2b).

Breast cancer, as a heterogeneous disease, has five well-established subtypes, including LumA, LumB, Her2 +, Basal-like, and Normal-like, that show distinct molecular profiles and different underlying mechanisms. We therefore checked if 1030 ncRNAs were significantly mutated in the five well-established breast cancer subtypes. We obtained the PAM50 subtype annotation of 346 ICGC breast cancer samples from a publication by Nik-Zainal et al.[30]. Of 1030 significant ncRNAs, we identified 782 ncRNAs mutated in these samples. Supplementary fig. S1 shows that most ncRNAs were mutated in multiple subtypes. However, we observed that each subtype also has its unique set of ncRNAs that were not mutated in other subtypes (see more details in Supplementary Data 2).

We then checked if the mutations in the candidate set of ncRNAs are breast cancer-specific or if they are also significantly mutated in other cancers. We first identified the candidate set of ncRNAs in 17 other cancer types. We then calculated how many of the 1030 significant ncRNAs in breast cancer are also significant in other cancers. As Table 1 shows, at least 427 of 1030 ncRNAs are explicitly mutated in breast cancer samples.

To see if the 1030 BC-associated ncRNAs are more BC specific or not, we sorted the significantly mutated ncRNAs in each cancer based on their mutational P value. We took 1030 top ncRNAs to see how many of them are common with the 1030 BC-associated ncRNAs. As Table 2 shows, at least 916 (%80) of the BC-associated ncRNAs were mutated explicitly in breast

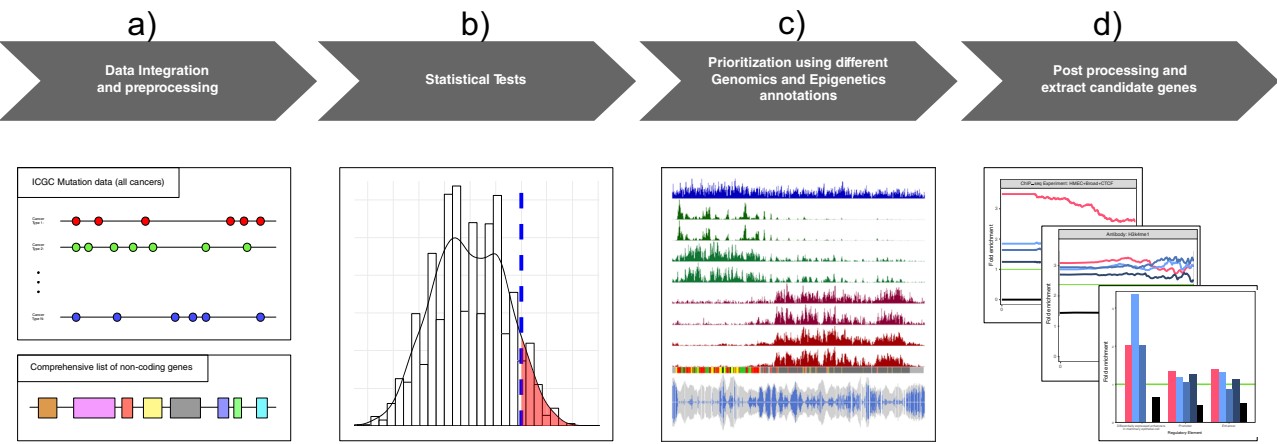

**Fig. 1 The flow diagram of the SomaGene pipeline is used in this study. a** ICGC cancer samples are used to identify BC-associated non-coding RNAs. Only samples with single point mutations are considered in the analysis. We also used a combined FANTOM5 robust gene list, Ensembl gene list, and the RNA Atlas[28] to have a comprehensive list of non-coding RNAs. **b** After counting the number of mutated samples in each ncRNA, we use Fisher's exact test, as described in the method section, to identify the mutational *P* value for each ncRNA. To identify significantly mutated ncRNAs, we calculate *P* values for 1,000,000 random permutations of breast/non-breast labels to estimate the 99% C.I. threshold of *P* value for each ncRNA. **c** We then investigate the overlapping of non-coding RNAs with breast tissue-related regulatory features (e.g., ENCODE predicted chromHMM, H3K27ac), BC-related GWAS SNPs, HMEC related eQTL polymorphisms, and HMEC related Hi-C interacting regions. **d** Finally, we provide a list of breast cancer-associated ncRNAs with potential enhancer activity.

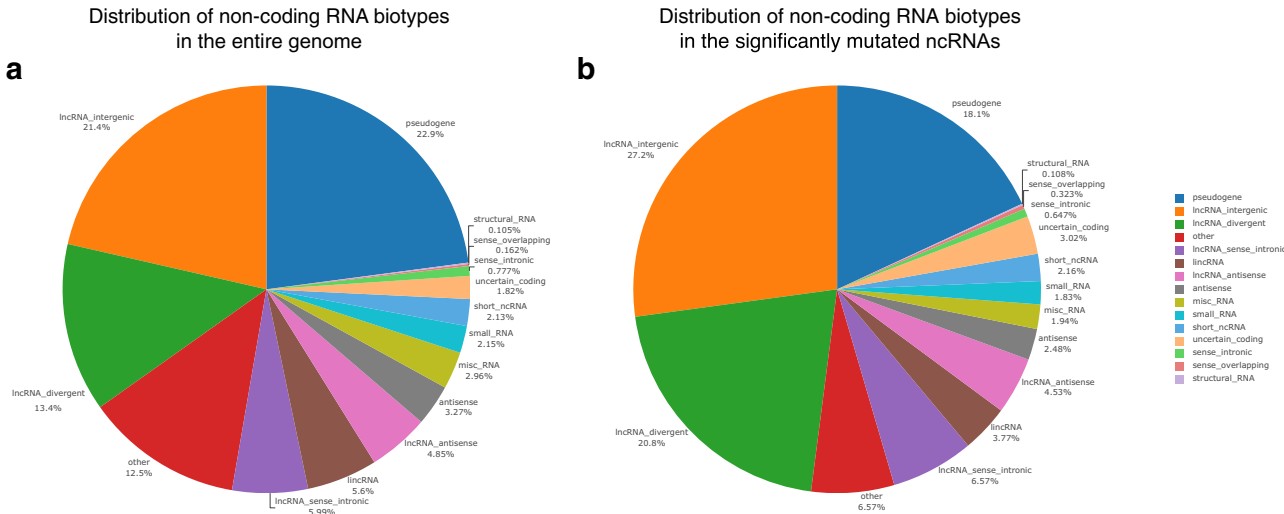

**Fig. 2 The distribution of non-coding RNA biotypes. a** For the entire genome. **b** For the set of significantly mutated non-coding RNAs.

cancers samples, indicating the 1030 ncRNAs identified in our study are more relevant to breast cancer than other cancers (e.g., more breast cancer-specific).

We next evaluated if 1030 significant ncRNAs were expressed ubiquitously across all breast cancer patients or if they were expressed in specific subtypes only. We used the expression dataset from Breast invasive carcinoma (BRCA) gene expression from TANRIC[31]. We found 504 ncRNAs of the 1030 significant ncRNAs in the TANRIC gene expression list. Of these, 106 ncRNAs were differentially expressed between the breast cancer subtypes. Supplementary fig. S2 showed that these 106 ncRNAs show some breast cancer subtype specificity. A list of differentially expressed ncRNAs is provided in Supplementary Data 3.

**BC-associated GWAS SNPs are significantly enriched in the candidate non-coding RNAs.** Multiple genome-wide association studies identified disease-associated genes and their respective pathways, which provided a comprehensive understanding of the disease's etiology. It has been reported that more than 93% of

disease-associated variations found by GWAS are located in the non-coding regulatory regions of genomes[32], suggesting non-coding regulatory regions are relevant to disease and genetic mutations in gene regulatory regions is a significant mechanical contributor to diseases. To examine the enrichment of BC-associated GWAS SNPs in the candidate ncRNAs, we extracted the BC-associated SNPs from a pooled list of two GWAS datasets from the EBI GWAS Catalog[33] and GWASdb v2 from the Wang Lab[34] (*for more details see the method section*). As Fig. 4a shows, BC-associated GWAS SNPs are significantly enriched (*P* value 2.3e−27) in the candidate ncRNAs. This enrichment is much higher for those ncRNAs that contain more than 4 GWAS SNPs (>10 times enrichment). Interestingly, when we performed the enrichment analysis for ncRNAs with more than 5 GWAS SNPs, only the candidate list of ncRNAs showed enrichment for GWAS SNPs (>20 times), and there was no enrichment in the 2nd and 3rd sets of ncRNAs (Fig. 4a). Performing the same analysis on lung cancer-associated GWAS SNPs did not show such enrichment for our candidate ncRNAs (Fig. 4a), indicating that

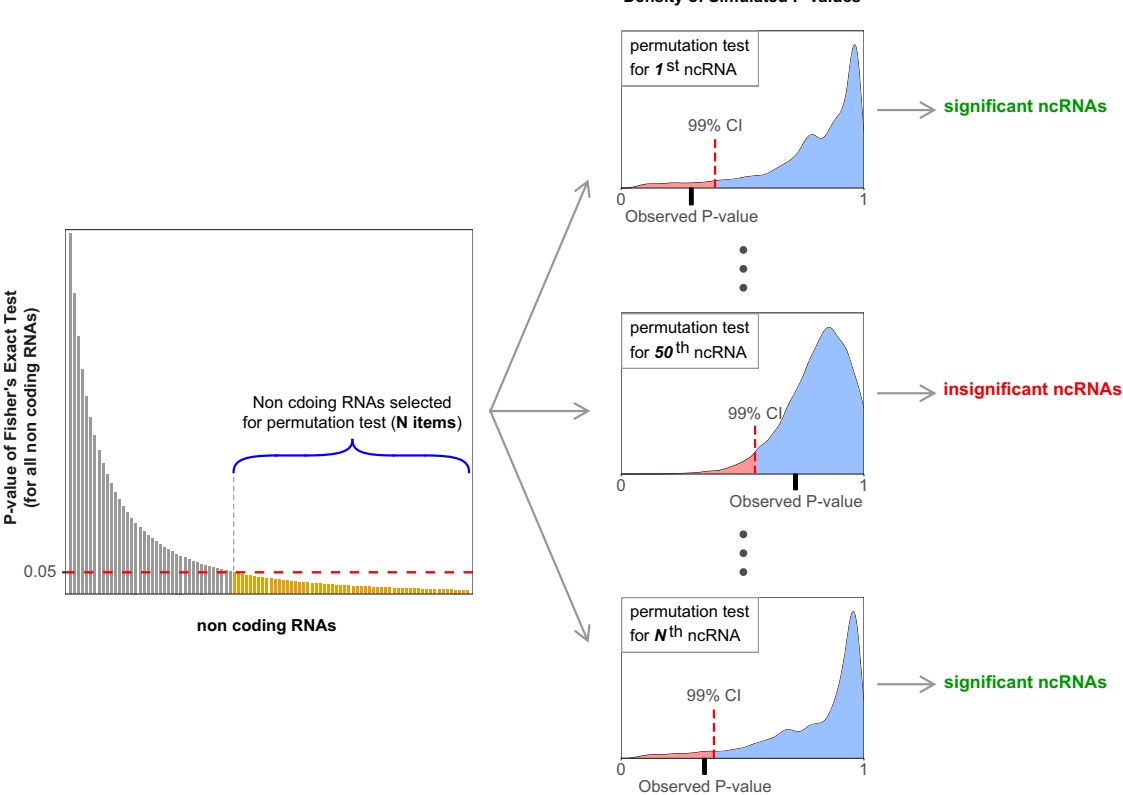

**Fig. 3 A schematic of permutation process to identify significantly mutated non-coding RNAs in breast cancer.** To determine the significance level of mutations in ncRNAs, we calculated $P$ values for 1,000,000 random permutations of breast/non-breast labels to estimate the 99% C.I. threshold of $P$ values. The permutation accounts for statistical significance based upon our original data sampling and avoids Type II errors that may arise from multiple testing correction approaches such as Bonferroni. We performed the permutations for each ncRNA separately. The red color indicates the 99% C.I. of observed $P$ values in the permutations in the figure.

candidate ncRNAs are relevant to breast cancer. For example, candidate ncRNAs *RP11-353N4.6* are known to carry breast cancer-associated GWAS SNPs[35]. More details on the annotated list of ncRNAs with BC-related GWAS SNPs can be found in Supplementary Data 4.

**BC-associated non-coding RNAs have a significantly higher fraction of eQTL polymorphisms**. eQTL are genomic locations that explain which genetic variations may change the gene function in a relevant tissue. To assess eQTL polymorphisms in the candidate non-coding genes, we calculated the enrichment of breast mammary tissue eQTL polymorphisms downloaded from GTEx consortia[36] in the candidate ncRNAs. Our analysis revealed that breast mammary tissue eQTL polymorphisms are significantly enriched (1.77 times with a $P$ value of 4.11e−20) in the candidate ncRNAs. Interestingly, this enrichment is much higher for those ncRNAs that contain more than three eQTL polymorphisms (>2× enrichment), and such an increase has not been seen in the 2nd and 3rd sets of ncRNAs (Fig. 4b and Supplementary Data 4). We calculated the enrichment of lung-specific eQTL polymorphisms in the candidate ncRNAs as a control. As Fig. 4b shows, the enrichment of lung-specific eQTL polymorphisms in the breast-associated ncRNAs is much lower than the enrichment observed for mammary tissue eQTL polymorphisms (Fig. 4b). For example, lncRNA *RP11-37B2.1* and *RP11-426C22.5* are two of our candidate ncRNAs with significant $P$ values of 7.91e−4 and 6.72e−06, respectively. These ncRNAs encompass 232 and 111 eQTL polymorphisms, respectively. lncRNA *RP11-37B2.1* influences the risk of tuberculosis and the possible correlation with adverse drug reactions from tuberculosis

treatment[37], and *RP11-426C22.5* is downregulated in SW1990/GZ Cells[38]. Both *RP11-37B2.1* and *RP11-426C22.5* overlapped with histone active mark H3K27ac, ChromHMM potent enhancer, and DHS, suggesting a potential transcription regulation role of these ncRNAs.

**Enrichment of promoter and enhancer signals in the BC-associated ncRNAs**. To determine if the somatic mutations are enriched in ncRNAs with enhancer activity, we first examined the enrichment of HMEC-related chromatin states provided by the ENCODE consortium within our significant ncRNAs. As Fig. 5a shows, both ENCODE promoters and enhancers have been significantly enriched within our candidate ncRNA genes (3.23 and 2.26 times with $P$ values 4.12e−29 and 5.28e−32 for promoters and enhancers, respectively), suggesting breast cancer de novo somatic mutations are enriched in ncRNAs with enhancer and/or promoter like functions.

We also investigated these enrichments in the 2nd and 3rd sets of ncRNAs (each set contains 1030 ncRNAs) and the last set of ncRNAs with worse mutational $P$ values (*see method section*). As Fig. 5a shows, the same enrichment trend (but much lower) for ChromHMM predicted promoters and enhancers in the 2nd and 3rd sets of most mutated ncRNAs in breast cancer (Fig. 5a). Interestingly, there is no such trend for the last set of ncRNAs (those with no de novo mutation in breast cancer samples), supporting our hypothesis that de novo somatic mutations are enriched in enhancer-like ncRNAs. We provided an annotated list of candidate ncRNAs with ChromHMM in Supplementary Data 5.

**Table 1 Number of significantly mutated ncRNAs in each cancer that are common with BC-associated ncRNAs.**

| Bladder | Blood | Bone | Brain | Breast | Cervix | Colorectal | Esophagus | Kidney | Liver | Lung | Ovary | Pancreas | Prostate | Skin | Stomach | Uterus |
|---|---|---|---|---|---|---|---|---|---|---|---|---|---|---|---|---|
| 3 | 94 | 0 | 0 | 1030 | 3 | 7 | 277 | 1 | 176 | 79 | 263 | 230 | 91 | 603 | 3 | 16 |

Each number in this table shows how many of the 1030 significant ncRNAs in breast cancer are also significantly mutated in other cancers.

**Table 2 Number of Top 1030 significant ncRNAs in each cancer that are common with BC-associated ncRNAs.**

| Bladder | Blood | Bone | Brain | Breast | Cervix | Colorectal | Esophagus | Kidney | Liver | Lung | Ovary | Pancreas | Prostate | Skin | Stomach | Uterus |
|---|---|---|---|---|---|---|---|---|---|---|---|---|---|---|---|---|
| 6 | 58 | 12 | 0 | 1030 | 5 | 1 | 6 | 1 | 17 | 30 | 116 | 97 | 87 | 8 | 3 | 20 |

Each number in this table shows how many of the top 1030 significant ncRNAs in other cancers are in the BC-associated ncRNAs.

The FANTOM5 consortium has released lists of human transcribed human promoters and tissue-specific transcribed enhancers of humans using CAGE (Cap Analysis of Gene Expression[39]) to study cell-type-specific enhancers. Therefore, we investigated the enrichment of FANTOM5 promoters and enhancers that overlap with the significant ncRNAs identified in this study. Figure 5b shows that both FANTOM5 promoters and enhancers are enriched in the candidate ncRNAs (1.66 and 1.76 times ($P$ value 3.59e−34 and 3.68e−27) enrichment for FANTOM5 promoters and enhancers, respectively). In other words, 52.4% of candidate ncRNAs overlapped with FANTOM5 promoters, and 36.4% of them overlapped with FANTOM5 enhancers. However, only 34.9% and 23.5% of all ncRNAs overlapped with FANTOM5 promoters and enhancers. The same analysis on FANTOM5 mammary-specific enhancers demonstrated that the proportion of candidate ncRNAs that overlap with differentially expressed enhancers in the mammary epithelial cell is 3.84 times ($P$ value 4.03e−05) more than the genome-wide expectation (Fig. 5b). There is also the same trend for the 2nd and 3rd sets of ncRNAs with the best mutational $P$ values in breast cancer. An annotated list of candidate ncRNAs with FANTOM5 annotations is provided in Supplementary Data 6. We also provided a list of candidate ncRNAs that overlap with ENCODE and FANTOM5 enhancer/promoter features in Supplementary Data 7. Notably, 317 ncRNAs overlapped with ENCODE and FANTOM5 enhancer marks; 257 ncRNAs overlapped with ENCODE and FANTOM5 promoters. For example, the pseudogene *NKAPP1* is differentially expressed in ABL1/ABL2 knockdown (shAA) breast cancer-associated cell lines[40] and downregulated in breast cancer[41]. It is also a biomarker associated with breast cancer prognosis[42,43]. Our analysis demonstrated that *NKAPP1* is one of the most significant ncRNAs, with a $P$ value of 3.43e−06. This non-coding gene overlapped with both FANTOM5 enhancer and ChromHMM predicted enhancer and promoter. *CATG00000062386* is another ncRNA gene that is significantly mutated in breast cancer samples. This FANTOM-CAT specific ncRNA overlapped with FANTOM5 and ChromHMM enhancers and FANTOM5 mammary epithelial cell differentially expressed enhancers, indicating a potential enhancer role for this ncRNA in breast cancer.

**Histone modifications H3K27ac and H3K4me1, CTCF binding sites, and DHS are significantly enriched for BC-associated ncRNAs.** H3K27ac and H3K4me1 are histone marks present at enhancer or promoter regions. DHSs are also known as the generic markers of regulatory DNA, containing genetic variations associated with diseases[44,45]. In addition to these marks, CTCF binding sites also have a wide-range regulatory function in the genome, in which mutations that reside in these regions affect the binding specificity to DNA sequences and may lead to aberrant expression of cancer-related genes[46]. To check if these important regulatory features are enriched in the candidate ncRNAs, we investigated the enrichment of HMEC-specific chromatin histone active marks (e.g., H3K27ac and H3K4me1) CTCF binding sites and DHS in the candidate ncRNAs identified in this study. These active chromatin marks are involved in many processes, including transcriptional regulation that regulates gene expression[47]. Our investigation of histone active marks demonstrated that both histone marks H3K27ac and H3K4me1 are significantly enriched 2.1 times ($P$ value 3.65e−57) and 1.6 times (1.90e−42) in the candidate ncRNAs (Fig. 6a). As these histone marks, suggesting our candidate list of ncRNAs is important for the transcriptional process in breast tissue. We performed the same analysis on histone marks H3K27me3, which is involved in the repression of transcription. Interestingly, we did not see significant enrichment

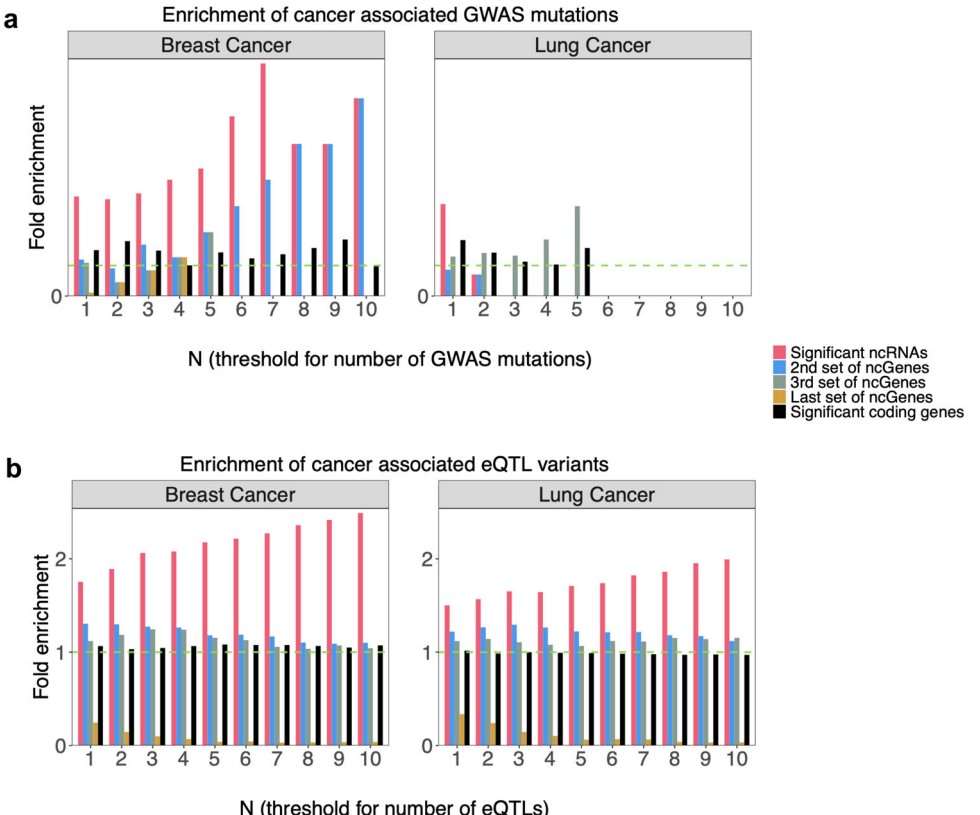

**Fig. 4 Enrichment of GWAS SNPs and eQTL polymorphisms in the significant set of ncRNAs. a** Enrichment of breast and lung cancer-associated GWAS SNPs in the candidate non-coding RNAs. **b** Enrichment of breast and lung tissues associated eQTL pairs in the candidate non-coding RNAs. We repeated the enrichment analysis with many items (e.g., GWAS SNP or eQTL polymorphisms) overlapping the ncRNAs. E.g., counting the number of ncRNAs that encompass at least 2/3/4/5/6/7/8/9/ GWAS SNPs or eQTL polymorphisms. As the figure shows, the candidate set of ncRNAs significantly enriched for both BC-related GWAS SNPs and breast tissue-related eQTL polymorphisms. This enrichment is much higher than the enrichment in lung cancer-related GWAS SNPs and lung tissue-related eQTL polymorphisms.

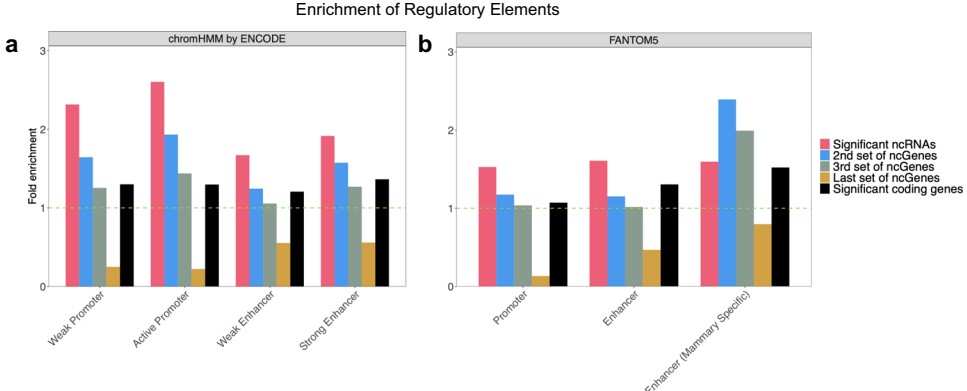

**Fig. 5 Enrichment of promoters and enhancers in the significant set of ncRNAs. a** Enrichment of HMEC-related promoters and enhancers identified by ENCODE (using chromatin segmentation by Hidden Markov Model (HMM). **b** Enrichment of promoters, enhancers, and breast tissue differentially expressed enhancers identified by FANTOM5 consortia. The enrichment is calculated by dividing the proportion of significantly mutated ncRNAs that overlap with each item by the proportion of all ncRNAs that overlap with that item. This enrichment is calculated for a significant set of ncRNAs (1030 ncRNAs) shown in red color, 2nd set (blue), 3rd set (gray) of highly mutated ncRNAs. The enrichment was also calculated for the last set of ncRNAs (brown) with a mutational *P* value close to 1. In other words, the last set refers to ncRNAs that had no mutation in breast cancer samples. Each set of ncRNAs contains 1030 elements.

(1.15 times with a *P* value of 5.14e−02) for histone H3K27me3 within our BC-associated ncRNAs (Fig. 6a).

Transcription factors CTCF function as a transcriptional activator, repressor, insulator, or pausing transcription. In addition to CTCF sites, DHS also has key roles in gene regulation

as regulatory element markers[48]. Both CTCF and DNase are functionally related to transcriptional activity and are necessary to regulate chromatin structure. Here, we choose three CTCF ChIP-seq experiments (HMEC + Broad + CTCF, HMEC + Broad + EZH2 and HMEC + UW + CTCF) and three DHS ChIP-seq

**a**

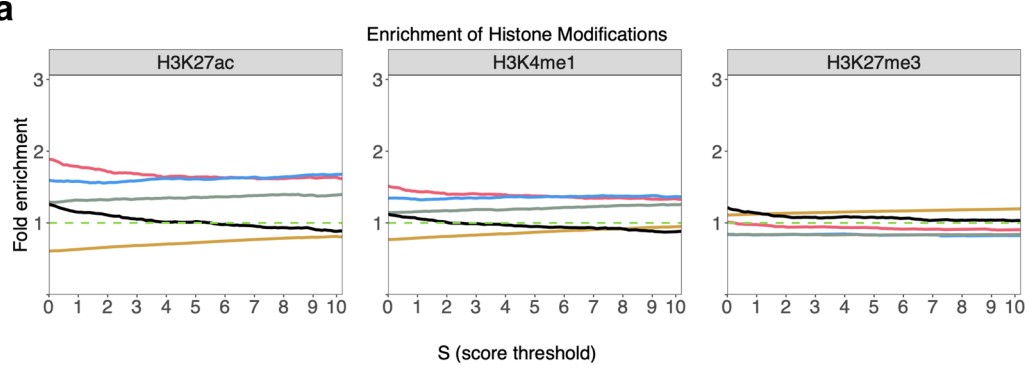

**b**

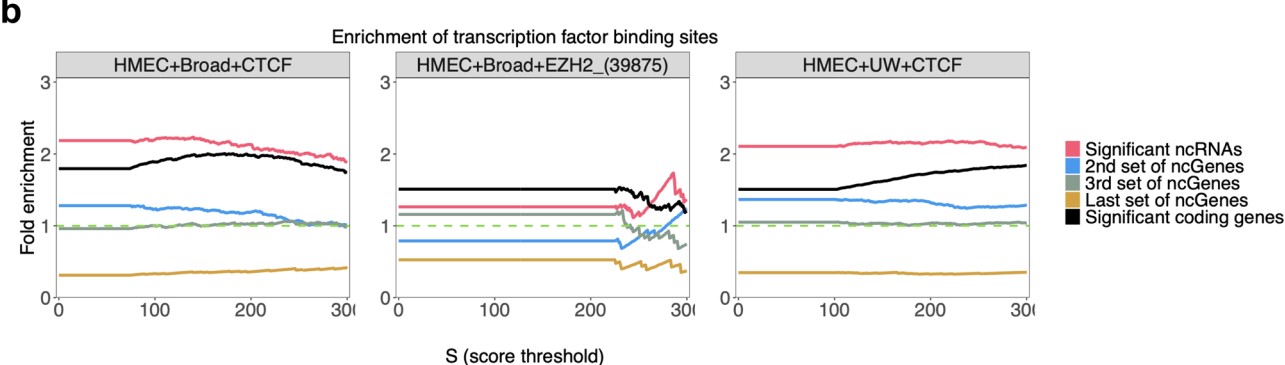

**c**

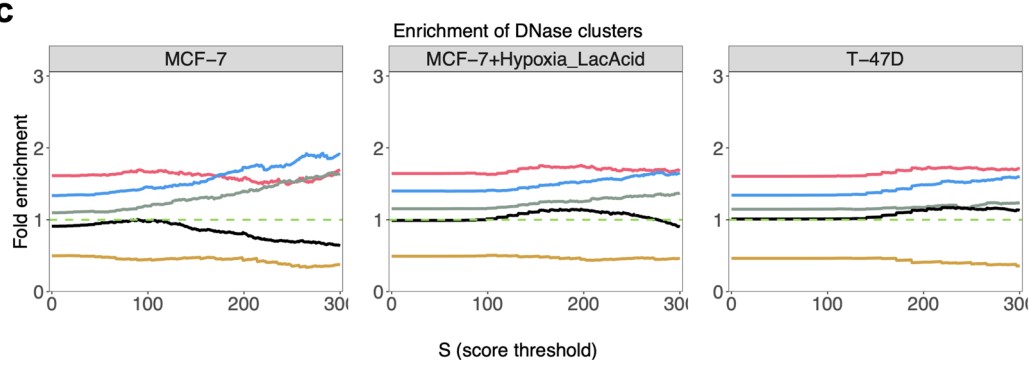

**Fig. 6 Enrichment of histone chromatin marks, transcription factor binding sites, and DNase clusters in the significant set of ncRNAs. a** Enrichment of histone modifications (for three antibodies: H3K27ac, H3K4me1, and H3K27me3). The candidate set of ncRNAs shows significant enrichment for H3K27ac and H3K4me1 and no significant enrichment for H3K27me3, which is a repressor histone mark. **b** Enrichment of transcription factor binding sites (for 3 ChIP-seq experiments: HMEC + Broad+CTCF, HMEC + Broad+EZH2 and HMEC + UW + CTCF). In all experiments, the candidate set shows significant enrichment, much higher than other sets of ncRNAs. **c** Enrichment of DNase clusters (for three cell types: MCF − 7, MCF − 7+Hypoxia_LacAcid, and T − 47D). As the figure shows, the candidate set of ncRNAs significantly enriched for DNase clusters, much higher for most of the scores than other sets. The enrichment is calculated by dividing the proportion of significantly mutated ncRNAs that overlap with each item by the proportion of all ncRNAs that overlap with the item. This enrichment is calculated for a significant set of ncRNAs (1030 ncRNAs) as shown in red color, 2nd set (blue), and 3rd set (gray) of highly mutated ncRNAs. The enrichment was also calculated for the last set of ncRNAs (brown) that had no mutation in breast cancers. Each set of ncRNAs contains 1030 elements. Note: The score threshold is the minimum threshold for signal-value when counting the number of overlaps. See the method section for more details.

experiments (MCF-7, MCF-7+Hypoxia_LacAcid, T-47D), all related to the breast cancer. We calculated the enrichment for the four sets of ncRNAs, including significant, 2nd, and 3rd sets of ncRNAs with the best mutational $P$ values and the last set of ncRNAs with the lowest mutational $P$ values. Our assessment of CTCF binding sites and DHS demonstrated that both CTCF (2.45 times, on average with $P$ value 7.42e−35) and DNase (1.8 times, on average with $P$ value 7.23e−50) are significantly enriched for our candidate ncRNA genes (Fig. 6b). Having such significant enrichment for CTCF binding sites and DNase accessible sites is

strong evidence that our significant set is not chosen randomly and is related to the gene regulation process. There is also the same trend for the 2nd and 3rd sets of ncRNAs (Fig. 6b). However, the enrichments for the candidate set are much higher than the 2nd and 3rd sets. As Fig. 6b shows, there is no enrichment for the last set of ncRNAs, suggesting that these ncRNAs may not be involved in transcriptional regulation.

For example, *LINC00535* is an antisense non-coding gene that is known to be associated with breast cancer[49]. *LINC00894* is another non-coding gene that is the most downregulated lncRNA

in MCF-7/TamR cells[50]. These ncRNAs are significantly mutated in breast cancer samples with *P* values 4.21e−3 and 1.53e−11. *LINC00152* is another example of ncRNAs with a substantial role in enhancing breast cancer, which causes inactivation of the BRCA1/PTEN by DNA methyltransferases as tumorigenesis, mainly in triple-negative breast cancer (TNBC)[51]. These ncRNAs overlapped with ENCODE-predicted enhancers, histone 27 acetylation, CTCF binding sites, and DHS, suggesting a potential transcriptional regulatory role for these ncRNAs. An annotated list of candidate ncRNAs with these features is provided in Supplementary Data 8 and 9.

**Chromosome conformation capture data shows a potential regulatory role for genomic loci that overlap with BC-associated ncRNAs.** High-throughput chromosome conformation capture (Hi-C) based assays have been used to successfully identify regulatory regions and targets of disease-associated variations[52,53]. To further understand the genes in which the genomic loci encompassing candidate ncRNAs interact, we analyzed two publicly available Hi-C datasets from HMEC cell lines[54]. We used MHiC[55] and MaxHiC[56] to analyze Hi-C raw data and identify statistically significant interactions. We identified 188,982 statistically significant interactions (*P* value < 0.01 and read-count ≥10—*see method section*) in the Hi-C library 1. For 6187 of the interactions (%3.3), one side of the interaction overlapped with at least one candidate ncRNA genomic region. Another side of the interaction overlapped with protein-coding genes (promoter regions of coding genes—*see method section*; Supplementary Data 10). Repeating this analysis on the second Hi-C library also identified 318,034 statistically significant interactions. For 9879 of the interactions (3.1%), one end of the interactions overlapped with the genomic region of candidate ncRNAs, and another end overlapped with the promoter region of protein-coding genes (Supplementary Data 10).

We identified 1167 common significant interactions between the two libraries where one side of the interaction overlaps with genomic regions that encompassed at least one candidate ncRNAs (Supplementary Data 11) and another side with protein-coding genes. In other words, for 251 ncRNAs (the genomic regions containing 251 ncRNAs) out of a pool of 1030 candidate ncRNAs (24%), there was at least one significant interaction in both the libraries (Supplementary Data 11); this is significantly higher (1.74 times; *P* value 4.61e−11) than genomic regions of all ncRNAs that overlapped with one side of the interactions in both the libraries (14%). The overlapping between 251 ncRNAs and regulatory features used in this study is shown in Supplementary fig. S3.

We then repeated the enrichment analyses to see if the 251 ncRNAs supported by multiple Hi-C-based assays have better enrichment of regulatory features, GWAS, and eQTL polymorphisms than other sets of ncRNAs. As Supplementary fig. S4 shows, the 251 ncRNAs have much higher enrichment of overlapping with regulatory features, GWAS, and eQTL polymorphisms than the remaining set of ncRNAs in the candidate list (1030-251 candidate ncRNAs), as well as than the ncRNAs in the 2nd, 3rd, and last sets of ncRNAs supported by multiple Hi-C based assays (Supplementary fig. S4). This supports our hypothesis that De novo somatic point mutations are enriched in enhancer-like ncRNAs. The list of the 251 ncRNAs is provided in Supplementary Data 11.

For 757 significant ncRNAs (%82), we identified at least one interaction in either Hi-C library 1 or library 2, resulting in 21,564 interactions. For 19,674 out of 21,564 interactions (Supplementary Data 12), one end of the interaction that encompasses candidate ncRNAs (genomic regions) overlapped with either ENCODE HMM predicted enhancer or active histone

mark H3K27ac (both presented in HMEC). This observation suggests a potential enhancer role for these ncRNAs; In many cases, another end of the interactions overlaps with protein-coding genes, including cancer-associated genes (Supplementary Data 12). We provided a prioritized list of candidate ncRNAs that interacted with cancer-associated protein-coding genes in the Hi-C libraries and overlapped with ENCODE HMM predicted enhancers and active histone mark H3K27ac (Table 3). For example, *MYC is* a BC-associated protein-coding gene acting as a transcription factor. In common with three other transcription factors (*POU5F1*, *SOX2*, and *KLF4*), it can induce epigenetic reprogramming of somatic cells to an embryonic pluripotent state[57]. In both Hi-C libraries, we found significant Hi-C interactions between *MYC* and two of our candidate ncRNAs, *CASC8* and *PVT1*. *PVT1* is a known enhancer for *MYC*[58]. However, *CASC8* has not yet been identified as a putative enhancer for *MYC*. There are ~200 kb genomic distances between the *CASC8* and transcription start site of *MYC* in which there is no protein-coding gene that overlaps with *CASC8*. There are strong signals of histone active marks and ENCODE predicted enhancers, and most importantly, a FANTOM5 breast differentially expressed enhancer overlapped with *CASC8*, suggesting a potential enhancer region in *CASC8* (Fig. 7). Interestingly, there is no somatic mutation or BC-associated GWAS SNPs that overlap with *MYC*. However, *CASC8* is significantly mutated in breast cancer samples, and most importantly, it encompasses 10 BC-associated GWAS SNPs. Altogether, this evidence may indicate a putative enhancer role of *CASC8* for *MYC*.

Another example is ncRNA *CATG00000061359*, a FANTOM5-specific intergenic lncRNA significantly mutated in breast cancer samples (*P* value 5.32e−4). There is a significant Hi-C interaction between *CATG00000061359* and gene *GTPBP8* (a known breast cancer-associated gene[59]) in both the libraries. We also found a breast tissue-associated eQTL polymorphism in this lncRNA that influences the expression of *GTPBP8* in breast tissue. Interestingly, both HMEC specific H3K27ac and HMM predicted enhancers overlap with this lncRNA. This suggests that variation in *CATG00000061359* may influence the expression of *GTPBP8* in breast cancer. We have provided an annotated list of candidate ncRNAs with Hi-C interactions in Supplementary Data 12.

## Discussion
Somatic point mutations play a key role in tumorigenesis and the development of cancer[60]. Recent studies on somatic mutation evolution in cancer have identified cancer driver genes[61] and mutational cancer signatures[62–64]. However, the analysis of somatic mutations has focused mainly on the protein-coding genes of the genome, and their potential impact on the non-coding RNA genes has been far less studied. ncRNAs have long been considered a non-functional part of the human genome[65]. However, these non-coding elements (majority lncRNAs) have recently opened a new insight into the study of breast cancer, acting as indispensable contributors to cellular activities, including the proliferation, apoptosis, survival, differentiation, and breast cancer metastasis[66]. In addition, ncRNAs have been used as biomarkers in many cancers, including breast cancer, through various mechanisms, including regulating the expression of protein-coding genes and functions at transcriptional, translational, and post-translational levels[21–25]. This indicates that ncRNAs may have the potential for diagnosis, prognosis, and therapeutics of cancers. This study identified ncRNAs that were mutated explicitly in breast cancer patients and then uncovered the connection between somatic point mutations in BC-associated ncRNAs and ncRNA regulatory properties in breast cancer.

**Table 3 Prioritized list of candidate ncRNAs and their linked protein-coding genes through a Hi-C interaction.**

| ncRNA interacting with PGCs | ncRNA mutational $P$ value | Protein-coding gene | Distance between ncRNA and PGC (bp) | PUBMED ID |
|---|---|---|---|---|
| LINC00894 | 1.49E−11 | MAMLD1 | 385,000 | PMID: 21559465 |
| MIR223 | 6.55E−10 | MSN | 250,000 | PMID: 9706140 |
| RNVU1-19 | 1.79E−06 | HIST2H2BF | 380,000 | |
| RNVU1-19 | 1.79E−06 | FCGR1A | 380,000 | DOI: 10.21203/rs.3.rs-38062/v1 |
| RNVU1-19 | 1.79E−06 | CATG00000015899.1 | 4,665,000 | |
| RP11-403I13.5 | 2.11E−06 | HIST2H2BF | 380,000 | |
| RP11-403I13.5 | 2.11E−06 | FCGR1A | 380,000 | DOI: 10.21203/rs.3.rs-38062/v1 |
| RP11-403I13.5 | 2.11E−06 | CATG00000015899.1 | 4,665,000 | |
| CATG00000028030.1 | 5.47E−06 | SWT1 | 395,000 | PMID: 16698800 |
| CATG00000028030.1 | 5.47E−06 | IVNS1ABP | 370,000 | |
| CASC8 RP11-96B2.1 | 1.28E−05 | TBC1D31 | 390,000 | |
| RP11-318M2.2 | 1.54E−05 | ATP6V1C1 | 165,000 | PMID: 24155661 |
| RP11-318M2.2 | 1.54E−05 | BAALC | 165,000 | PMID: 12750167 |
| GPX1P1 | 2.09E−05 | PRPS2 | 440,000 | PMID: 24855946 |
| GPX1P1 | 2.09E−05 | CATG00000113128.1 | 440,000 | |
| RP11-296O14.3 | 4.39E−05 | CENPL | 410,000 | |
| RP11-296O14.3 | 4.39E−05 | DARS2 | 410,000 | |
| AF196970.3 | 6.36E−05 | OTUD5 | 250,000 | PMID: 32655987 |
| AF196970.3 | 6.36E−05 | CATG00000111204.1 | 250,000 | |
| AF196970.3 | 6.36E−05 | KCND1 | 250,000 | PMID: 30051729 |
| RP1-15D23.2 | 8.89E−05 | TNFSF4 | 470,000 | PMID: 31501955 |
| RP1-15D23.2 | 8.89E−05 | PIGC | 280,000 | |
| RP1-15D23.2 | 8.89E−05 | C1orf105 | 280,000 | |
| RP1-15D23.2 | 8.89E−05 | SUCO | 130,000 | PMID: 31434866 |
| RP1-15D23.2 | 8.89E−05 | FASLG | 10,000 | PMID: 25394756 |
| RP11-426C22.5 | 1.16E−04 | BANP | 58,985,000 | PMID: 28103507 |
| RP11-973F15.1 | 1.28E−04 | TBC1D31 | 390,000 | |
| RP11-973F15.1 | 1.28E−04 | DERL1 | 295,000 | PMID: 20375427 |
| RP11-973F15.1 | 1.28E−04 | ZHX2 | 90,000 | PMID: 19273305 |
| CATG00000095444.1 | 1.82E−04 | MPP6 | 260,000 | PMID: 31402947 |
| CATG00000113234.1 | 2.09E−04 | REPS2 | 525,000 | PMID: 19776672 |
| CATG00000103306.1 | 2.79E−04 | PTDSS1 | 835,000 | |
| CATG00000103306.1 | 2.79E−04 | SDC2 | 555,000 | PMID: 23747112 |
| CATG00000103306.1 | 2.79E−04 | CPQ | 460,000 | |
| CATG00000017091.1 | 3.04E−04 | CATG00000015899.1 | 3,995,000 | |
| CATG00000017091.1 | 3.04E−04 | PDE4DIP | 3,455,000 | PMID: 30030436 |
| RP4-668J24.2 | 3.55E−04 | EXOC2 | 795,000 | |
| RP11-94A24.1 | 4.40E−04 | TBC1D31 | 390,000 | |
| RP1-60N8.1 | 4.88E−04 | REPS2 | 345,000 | PMID: 19776672 |
| RP11-177F15.1 | 4.97E−04 | ZP4 | 185,000 | |
| CATG00000083054 | 6.21E−04 | TNFSF4 | 380,000 | PMID: 31501955 |
| RP11-689K5.3 | 6.29E−04 | PRKG2 | 420,000 | |
| CATG00000087401.1 | 6.40E−04 | UCHL5 | 195,000 | PMID: 28681694 |
| CATG00000098967.1 | 6.45E−04 | IRF2BP2 | 380,000 | PMID: 23185413 |

Prioritized list of BC-associated ncRNAs that interact with protein-coding genes (PGCs) in both Hi-C libraries and overlapped with either ENCODE HMM predicted enhancer or active histone mark H3K27ac. A PUBMED ID is provided if the PGC is known to be associated with cancer. A detailed list of Hi-C data analyses is given in Supplementary Data 8 and 9.

As the previous study shown, tissue specificity is an important aspect of many genetic diseases, including cancers[67]; Our study demonstrated that the breast cancer-related ncRNAs have much higher enrichment of breast tissue/cell line-specific features; we have shown that the candidate ncRNAs with enrichment of somatic point mutations have a much higher fraction of regulatory features compared to genome-wide expectation, suggesting the potential impact of somatic mutations on the regulatory function of ncRNAs. Notably, we have shown that most of the candidate ncRNAs interact with promoters of protein-coding genes, again indicating the potential regulatory role of ncRNAs with significant enrichment of somatic point mutations in breast cancer.

Researchers have recently shown that germline mutations are associated with an increased risk of developing BC[68,69] and acquired somatic mutations driving the disease, in which germline mutations may interact with somatic mutations to drive carcinogenesis or involve in tumorigenesis by contributing to critical biological and cellular processes. Our analyses revealed that the ncRNAs with enhancer-like activity are significantly overlapped with breast cancer-associated GWAS SNPs and breast tissue-related eQTL polymorphisms. This highlights the importance of somatic and germline mutations in breast cancer development and progression.

The enrichment of somatic mutations in the ncRNAs with enhancer-like activity has not been previously explored. Our Hi-C

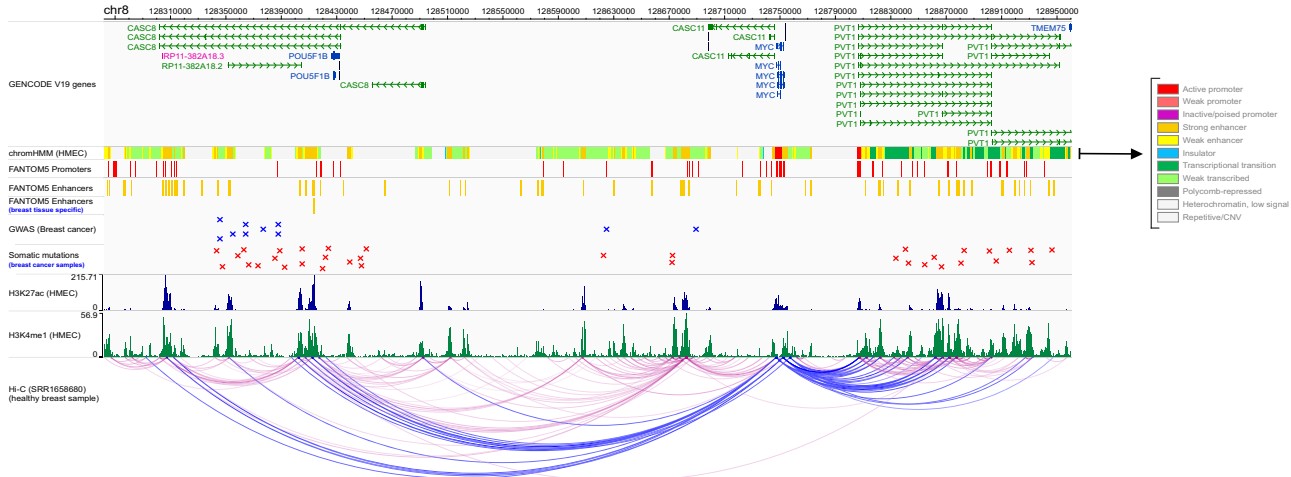

**Fig. 7 An example of Hi-C interaction between ncRNA *CASC8* and protein-coding gene *MYC*.** Long non-coding RNA *CASC8* is a breast tissue expressed lncRNA that is significantly mutated in breast cancer samples. Our analysis of HMEC-related Hi-C data shows that this lncRNA is significantly interacting with the promoter of multiple coding genes, including *MYC*, a known breast cancer-associated gene. There are numerous strong signals of ENCODE predicted ChromHMM potent enhancers, histone active marks H3K27ac, and H3K4me1 (all presented in HMEC) that overlap with *CASC8*. Notably, a FANTOM5 breast differentially expressed enhancer also overlapped with *CASC8*. Our analysis of GWAS SNPs and de Novo somatic point mutations revealed that *CASC8* covered multiple breast cancer GWAS SNPs and many somatic point mutations related to breast cancer samples. In contrast, *MYC* does not encompass either GWAS SNP or BC-related somatic mutations. The figure also shows that the *MYC* gene interacts with *PVT1*, another significantly mutated lncRNA in the breast cancer sample (30 kb far from *MYC*). *PVT1* is also overlapped with breast tissue-related regulatory features and is a previously known enhancer for the *MYC* gene. We used MHiC[55] to analyze raw Hi-C data and MaxHiC[56] to identify significant Hi-C interactions. Significant interactions are shown in blue color. ENCODE predicted chromHMM files chromatin signals were used in peak format.

analyses also provided more evidence of enhancer activity of genomic loci encompassing BC-associated ncRNAs. Interestingly, we showed that the 251 ncRNAs supported by multiple Hi-C-based assays have much higher enrichment of regulatory features, GWAS, and eQTL polymorphisms than the remaining ncRNAs in the candidate list, indicating these ncRNAs are more likely to be involved in the gene regulation process.

However, our analyses are limited to the associations identified by pure data-driven analyses. Future experimental validation can reveal finer details about the interdependence of ncRNAs function in breast cancer.

We have developed an extensive web-based resource to communicate our results with the research community. Further works could focus on the candidate ncRNAs provided in this resource to check their complex regulatory functions and reveal the novel mechanism underlying carcinogenesis and breast cancer treatment.

This study also presented a novel computational method, SomaGene, to prioritize and predict disease-associated ncRNAs by integrating multi-level omics data, including somatic mutations, transcriptomic and epigenetic signals, and chromatin conformation data.

## Methods

**Tool availability**. The SomaGene open-source R package, a sample dataset, and instructions on running SomaGene are provided at https://github.com/bcb-sut/SomaGene.

**ICGC dataset**. We used the ICGC dataset, which contains somatic point mutations from 1855 breast cancer samples and 10,419 samples from 17 different types of cancers.

**A combined list of non-coding RNAs from FANTOMCAT and Ensembl consortia**. We have combined two gene lists from FANTOM5[39] and Ensembl[70] consortia (genome building hg19), enabling us to have a comprehensive list of non-coding RNA genes. We used an in-house script to combine the lists based on gene coordinates and/or gene names. If both consortiums have the same gene but different gene coordinates, we considered the FANTOM5 genes as the priority. We also added a recently published list of long non-coding RNAs from an atlas of non-

coding RNAs[28] into our list of ncRNAs. In total, 65,257 non-coding RNA genes were available in the combined list.

**Identify significantly mutated non-coding genes**. Many computational techniques have recently been used for mutational information to investigate biomarkers in human and viral genomes[31,49,71–73]. In this study, we developed SomaGene, which uses a "vcf" file from the whole genome or exome data. We included all samples from ICGC with at least one de novo mutation. We only considered single nucleotide mutations and excluded insertions or deletions from the analyses. To identify ncRNAs that significantly mutated in breast cancer samples compared to other cancers, we used Fisher's exact test and permutation testing in the following manner:

We calculated a $P$ value for each ncRNA using a one-sided Fisher's exact test applied to a $2 \times 2$ contingency table whose elements are (1) the number of samples in breast cancer that are mutated in an ncRNA, (2) the number of samples in breast cancer that are not mutated in an ncRNA, (3) the number of samples in all cancers other than breast cancer that are mutated in an ncRNA and (4) the number of samples in all cancers other than breast cancer that are not mutated in an ncRNA (Supplementary fig. S5). To identify significant ncRNAs, we calculated $P$ values for 1,000,000 random permutations (which can be defined by the user) of sample IDs across all cancers to estimate the probability that an association emerges by chance at a confidence interval of 99%. We identified a total of 1030 significantly mutated ncRNAs in breast cancer.

**Overlap and aggregation score methods**. To adequately examine the overlapping of ncRNAs with regulatory features and calculate the overlapping score with each element, each annotation's overlapping ranges were used in our analysis. In the case of FANTOM5 promoters, enhancers, and tissue-specific enhancers, three binary variables for each ncRNA were calculated, indicating that an ncRNA has overlapped with any enhancer or promoter in the FANTOM5 dataset (Supplementary Data 6). In eQTL annotation, a set of all entries whose location overlaps with each ncRNA was extracted by concatenating the variation_id and gene_id for each location.

The chromatin segmentation annotation comprises genomic ranges, each attributed to one of 11 functional categories (segments). The percentage of overlap with a segment was calculated as the sum of proportions of nucleotides in that segment's ranges overlapped with the ncRNA. Thus, a profile of overlapping chromatin segments was calculated with their corresponding coverage over the ncRNA (Supplementary fig. S6, Supplementary Data 5).

For histone modifications annotation, besides the proportions of nucleotides covered with the overlapping histone modification ranges in each ncRNA, the peak scores of these ranges were averaged together by the corresponding overlap percentages as their weights to obtain a single histone modification score. The coverage (overlap percentage) of each ncRNA with histone modification ranges was

also calculated as the total proportion of nucleotides in the ncRNA covered with histone modification ranges.

In the case of DNase annotation, each genomic range is attributed to a group of cell types that show DNase hypersensitivity in that area. Several cell type IDs with corresponding DNase hypersensitivity scores are associated for each range. We combined the overlap annotation ranges to get an aggregated set of cell-type IDs, scores, and overlap percentages by calculating the proportion of nucleotides in the ncRNA covered with these ranges. After this step, several IDs were duplicated in many aggregated sets summed over the overlap percentages to obtain a single overlap measure. Also, it was averaged over its scores by the corresponding overlap percentages as their weights to take a single score for that ID. For each genomic range in transcription factors annotation, a group of transcription factors (from ENCODE) with their corresponding ChIP-Seq peaks are reported. While the schema of transcription factors annotation is similar to that of DNase annotation, the same procedure as described for DNase annotation was performed to obtain an aggregated set of transcription factor IDs, scores, and overlapping percentages for each ncRNA (Supplementary Data 8). The overlap of BC-related GWAS mutations loci with the overlapped ncRNA regions was identified. The total number of overlaps for each ncRNA was recorded in the output table (Supplementary Data 4).

**Calculating enrichments**. Every "enrichment" that is calculated throughout this study is defined as "the fraction of ncRNAs in the significant set that have the trait of interest (e.g., having overlap with or having a minimum score of a specific annotation) divided by genome-wide expectation." It can be easily justified that the mentioned value equals the fraction of items in the whole set of ncRNAs with the trait of interest. Let's assume that the total number of ncRNAs is $N$. The number of significant ncRNAs is $S$ while $A$ items among all and $Y$ items within the significant set have the property of interest. Suppose a random collection of ncRNAs with size $S$ is sampled. In that case, the probability that $X$ items within this set have the property of interest follows a binomial distribution with parameters $S$ and $A/N$, i.e., $X \sim Binom(S, A/N)$. Thus, the expected value of $X$ equals $S \times A/N$ which shows that the expected fraction of items having the property of interest in a random set of ncRNAs with size $S$ equals $A/N$. We then conclude that the defined enrichment can be calculated as $(Y/S)/(A/N)$.

**FANTOM5 promoters, enhancers, and breast differentially expressed enhancers**. We used the FANTOM5 CAGE expression atlas[26] to identify a set of significant ncRNAs that overlapped with FANTOM5 promoters and enhancers. The entire collection of enhancers and promoters found in the FANTOM5 data were downloaded from the FANTOM5 Phase2[39,74]. We also downloaded FANTOM5 breast differentially expressed enhancers from ref. [75].

**ENCODE chromatin state segmentation**. The Chromatin State Segmentation uses a standard set of states learned by computationally integrating ChIP-Seq data for nine factors plus input using a Hidden Markov Model (HMM) across various cell types. Also, it shows a classification of chromatin, like "enhancer," "promoter," or "repressed." We used this dataset to identify the set of breast-specific candidate ncRNAs that overlapped with chromatin states. The complete set of chromatin state segmentations for the models derived from HMECs, grown in vitro related to breast cancer was downloaded from the ENCODE project[8].

**ENCODE histone modifications by ChIP-Seq**. For this study, we used HMEC-specific ChIP-Seq data in processed peak calls for histone modifications H3K27ac, H3K27me1, H3K4me3, and CTCF from the ENCODE project to identify a set of significant ncRNAs that overlapped with three types of histone modifications[8].

**ENCODE transcription factor ChIP-Seq data**. This data shows regions of transcription factor binding derived from an extensive collection of ChIP-Seq experiments and DNA binding motifs identified within these regions by the ENCODE Factorbook repository[76]. The transcription factors are responsible for modulating gene transcription bind as assayed by chromatin immunoprecipitation with antibodies specific to the transcription factor followed by sequencing the precipitated DNA (ChIP-Seq). We used this dataset to identify a set of significant ncRNAs that overlapped with transcription factor ChIP-Seq. The set of transcription factor ChIP-Seq data for the HMEC cell line was downloaded from ENCODE project[8].

**DNase clusters**. Regulatory regions tend to be DNase-sensitive which are accessible chromatin zones and functionally related to transcriptional activity. We used DHS from MCF-7 and T47D from ENCODE project to identify a set of significant ncRNAs that overlapped with DNase clusters.

**Hi-C data analysis**. Hi-C is an experiment for identifying the number of interactions between genomic loci in a 3D space. Our study used two replications related to the HMEC[54]. We used MHiC[55] and Hi-C Pro[77] with the default parameters for analyzing and aligning Hi-C data in 5 kb fragment size. We used MaxHiC[56] as a background correction model to identify significant Hi-C interactions for true cis-interaction. Here, we included those significant interactions with $P$ value < 0.01, read-count ≥10, with the distance between two sides of interaction more than 5 kb and <20 Mb. We then annotated Hi-C interactions with coding and non-coding genes from our combined genes list. At least 10% overlap between gene and Hi-C fragments has been considered to annotate Hi-C fragments with genes.

**Genotype-tissue expression eQTLs**. eQTL data were downloaded from the Genotype-Tissue Expression (GTEx) Project[36]. We used GTEx v7 eQTLs identified as significant in the HMEC line from the GTEx project.

**Literature search for non-coding interacting genes**. Our literature searches were focused on human studies and English language publications available in PubMed, Scopus, and Web of Science. Both Medical Subject Headings terms and related free words were used to increase the sensitivity of the search. We also used data and text mining techniques to extract additional associated studies[78–85]. A decision tree approach and a knowledge-based filtering system technique have also been used to categorize the texts from the literature search[82,86,87]. The search terms included "non-coding RNA" or "lncRNAs" or "genes name + cancer." "BC" or "breast carcinoma" and "breast neoplasm."

**Genome-wide association study (GWAS) analysis**. We have pooled two GWAS datasets, EBI GWAS Catalog[33] and GWASdb v2, from Wang Lab[34] to identify a set of significant ncRNAs that overlapped with GWAS SNPs. We first converted all GWAS SNP coordinates to UCSC hg19 using UCSC Lift Genome Annotations tools[88] (www.ncbi.nlm.nih.gov/genome/tools/remap) and then used an in-house script to combine both GWAS datasets based on gene coordinates and/or gene symbols. All GWAS SNPs with $P$ value < e−8 were included in the analyses.

**Reporting summary**. Further information on research design is available in the Nature Research Reporting Summary linked to this article.

## Data availability

A sample dataset can be accessed at https://github.com/bcb-sut/SomaGene. The whole dataset is publicly accessible through ICGC data portal (https://dcc.icgc.org).

## Code availability

The source code can be accessed at https://github.com/bcb-sut/SomaGene.

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

## Acknowledgements

This work was funded by the UNSW Scientia Program Fellowship and the Australian Research Council Discovery Early Career Researcher Award (DECRA) under grant DE220101210 to H.A.R. We kindly acknowledge the Government of Western Australia, Department of Health, Clinical Excellence, for their kind support on this project through the MERIT award to H.A.R. H.A.R is also supported by UNSW Scientia Program Fellowship. Analysis was made possible with computational resources provided by the BioMedical Machine Learning Bioinformatics Server with funding from the Australian Government and the UNSW SYDNEY. H.R.R is supported by IRN National Science Foundation (INSF) Grant No. 96006077.

## Author contributions

H.A.R. designed the study; H.A.R., N.R., and M.B. wrote the paper. The paper was edited by H.A.R., N.R., J.B., M.S.T., M.B., N.H.L., and H.R.R. N.R., M.B., and M.H. carried out all the analyses, including the statistical analyses, gene prioritization, annotation, permutation, and Hi-C data analysis, with the supervision of H.A.R. and H.R.R. N.R. and M.B. generated all figures and all tables with the supervision of H.A.R. and H.R.R. S.K. designed and developed the website. All authors have read and approved the final version of the paper.

## Competing interests

The authors declare no competing interests. H.A-R. is an Editorial Board Member for Communications Biology but was not involved in the editorial review or decision to publish this paper.

## Ethics approval and consent to participate

The ethical approval was not needed; all the data used in this study are publicly available.
