## [Peer Review File · Communications Biology]

Reviewers' comments:

Reviewer #1 (Remarks to the Author):

The authors developed an R-based tool, dubbed as SomaGene, to identify 929 non-coding RNAs (ncRNAs) bearing significantly more somatic point mutations among >1800 patients diagnosed with breast carcinomas, compared to other types of cancer patients profiled by the ICGC. Through integrating multi-omics data including 1) chromatin states, histone modifications, TFBSs, and DNaseI hypersensitivity sites profiled by ENCODE, 2) actively transcribed promoters and enhancers identified by FANTOM5, 3) GWAS SNPs and eQTL polymorphisms, and 4) Hi-C assays for chromosome contact maps, the authors argued that these 929 ncRNAs have a great regulatory potential for protein-coding genes in breast cancers. In addition to breast tissue, the authors had deposited their analytical results for other five tissue types, including liver, blood, bone, lung, and skin, in their webtool to better share their findings with the ncRNA community. Despite their analysis pipelines showed little novelties and similar ideas can be found elsewhere, both standalone and online implementations displayed the results in a highly integrated manner which are beneficial to the ncRNA researchers interested in performing comprehensive analysis. My specific comments are listed below.

Major:

1) Breast cancer is a heterogeneous disease. Four well-established breast cancer subtypes (LumA, LumB, Her2+, and Basal-like) exhibited distinct molecular profiles and the underlying mechanisms driving the diseases are different between them. As ncRNAs are often expressed in a highly tissue- and context-specific fashion, I will expect each subtype has its unique set of ncRNAs significantly enriched for somatic point mutations. Please include this information to your analysis throughout the manuscript.

In addition, given the authors only studied somatic point mutations on the transcribed ncRNA transcripts rather than those across all non-coding DNA regions. Please examine if these 929 significant ncRNAs were expressed ubiquitously across all breast cancer patients or in certain subtypes only. If some of them were not expressed adequately, please reason how they exerted regulatory functions in breast cancer. Please also provide explanations why other mutations in non-coding DNA regions were excluded from your analysis.

2) An expanded ncRNA catalogue is available. A recent work had expanded the catalogue of human ncRNAs with >3300 previously uncharacterized lncRNAs through profiling 300 human tissues and cell lines using PolyA+ and Total RNA-Seq (PMID: 34140680). Please include these novel lncRNAs to your analysis for the most updated results.

Minor:

1) While calling significance, please compare ncRNAs with the protein-coding genes for the five sections in Pages 4-8. This will help your readers understand how well these ncRNAs are associated with breast cancers over their protein-coding counterparts.

2) For 226 ncRNAs supported by multiple Hi-C based assays, they should be separated from the significant set of 929 ncRNAs (or the 1st set) to see if they have better enrichment than other ncRNAs in the same set. In addition, please include the 2nd, 3rd, and last sets of ncRNAs in your Hi-C data analysis.

3) HMEC is a normal breast epithelial cell line. Please explain why MCF7 or other breast cancer cell lines profiled by ENCODE were not considered.

4) How many GWAS SNPs and eQTL polymorphisms were overlapped with the somatic point mutations? If a significant overlap is observed, please explain if the results in Pages 7-8 are essentially the product of circularity biases. Conversely, please discuss if both germline and somatic mutations are required to develop breast cancers.

Reviewer #2 (Remarks to the Author):

The authors have investigated the role of somatic mutation found within non-coding RNA genomic regions in breast cancer, and further investigate the association of mutated non-coding RNAs with several features, including annotated promoters/enhancers, histone modification, CTCF binding sites, DNase hypersensitive sites, GWAS SNPs, eQTL, and Hi-C. The manuscript includes several well performed in silico analysis and the method section is satisfactory written. The language is of good quality, but there are multiple sentences with poor quality that needs to be rewritten. Some of the figures needs also to be better presented. The peer reviewers major concern is that for the association analysis, most of these features included are associated with functional DNA, and less with functional RNA. These analyses are extremely interesting for identifying new DNA regulatory elements, but the association towards non-coding RNA transcripts is lacking. The manuscript therefor needs to be revised in order to better state how these analyses are relevant for non-coding RNAs. Some new analysis regarding the RNA expression needs to be performed, and the manuscript must in its introduction give an introduction towards the subtypes of enhancer and promoters non-coding RNAs. All of my concerns are more thoroughly addressed below.

Major concerns:

The expression of the non-coding RNAs in breast cancer has not been validated in this manuscript. Annotation of non-coding RNAs is still poor, and many are wrongly annotated. The authors should therefor examine the presence of these non-doing RNAs in transcriptome data from breast cancer patients and from normal breast tissue. Non-coding RNAs are known to be very tissue specific, and by examining if the non-coding RNAs are expressed in breast tissue and cancer, where there is an enrichment for mutation within these regions, one might get a better understanding if the mutations have an effect at the RNA or DNA level. If they are not expressed, the region might be important as a DNA enhancer/promoter region, versus a functional RNA. This is reflected by the analysis of non-coding RNAs that overlap with promoters/enhancers, where 52.4% of candidate ncRNAs overlapped with FANTOM5 promoters, and 36.4% of them overlapped with FANTOM5 enhancers. This might point toward the mutations identified being more important as for example a transcription factor binding site at the DNA level, and not being important in causing changes in a non-coding RNA. It is also known that single point mutation in non-coding RNAs can be less disruptive than for a protein coding gene, as the stability of the RNA might not be as affected by a single nucleotide change. The authors should for this reason also examine if know transcription factors have been identified to bind to the mutated regions.

In abstract line 10, the authors use of the word "explicitly". This word can make the reader think that the mutations are exclusively in breast cancer, which the peer reviewer find them not to be based on the presented data. The mutations were significantly when compared to an average of multiple other cancers, but this does not exclude that they could also have been significantly enriched in other specific type of cancer. The number of patients with a specific mutation found in a non-coding region, should be presented in a plot where the other cancers are separated into their individual subtypes, in order to check if the mutations were breast cancer specific, or were also specific to other cancer types. In line 71, the authors use the term specifically, which I feel is not thoroughly backed up based on how the comparison was done (compared against a pan-cancer average).

Figure 1 and the figure legend is not corresponding well. The figure should have included A,B,C, and D, in order to point to where in the figure the legend is referring to. The same concern is also for figure 2, where the figure is not easy to interpret from the figure legend, and does not give the reader any extra information regarding the results or better understanding of how the data was generated. The authors should carefully go through the figures and do modifications, as well as modify the figure legends so that they explain the figures and their data better. In figure 4, please explain the S (score threshold). Figure 3, 4, and 5 have the same figure heading in the figure legends.

The chapter "Enrichment of regulatory features in the BC-associated ncRNAs" might not have a good heading, as it suggests that multiple regulatory features will be examined in this chapter, while only enhancers/promoter regions are examined. It is also not clear from the first lines in the

introduction to this chapter what type of features they are examining, and the authors should state why they assign these types of regions as regions important for non-coding RNAs with regulatory functions. Also the 2nd and 3rd sets of ncRNAs should be better described when first introduced in this chapter. It would have been beneficial to include a figure showing the overlap between the numbers of non-coding RNAs identified by ENCODE and FANTOM (not only as a supplementary table), to see their overlap.

For the chapter "Histone modifications H3K27ac and H3K4me1, CTCF binding sites, and DNase hypersensitive sites are significantly enriched for BC-associated ncRNAs" the authors should give a more specific description regarding the markers and how these markers are relevant for non-coding RNAs. Here, it would be interesting to see the overlap with these features across each other and with the enhancer/promoter analysis performed in the chapter above, as this could point towards specific non-coding RNAs being highly associated with DNA regulatory regions. Personally, I would have been focusing more on the non-coding RNAs not being enriched by these features and the enhancers/promoters, as the features have a stronger association towards regulatory DNA regions than functional RNAs. Better description on how these features have previously been by the literature used to identify functional RNAs, would help the authors convince the readers about the relevance for conducting these types of correlation analysis.

The chapter "BC-associated GWAS SNPs are significantly enriched in the candidate non-coding RNAs" is more a confirmation of the results from the section "Background model to identify significant non-coding RNAs in breast cancer" and should be written together as a single chapter in the beginning of the manuscript, or come as an independent chapter right after the first chapter. Similarly, the chapter "BC-associated non-coding RNAs have a significantly higher fraction of eQTL polymorphisms" should be moved up as this also gives evidence towards these non-coding RNAs being important in disease.

The chapter "Chromosome conformation capture data shows a potential regulatory role for BC-associated non-coding RNAs" includes an analysis where the DNA is in focus, and where it is a lack in understanding how this analysis gives evidence for a functional RNA. Also for this analysis, it would be beneficial to present the overlap with the previous analysis in a figure. The proposed inclusion of expression analysis of the non-coding RNAs, could be supplemented with correlation analysis of expression values between the non-coding RNAs with their chromosomal interacting protein coding genes.

The statement "Our analysis demonstrated that highly mutated lncRNAs have more cell type-specific functions and are more likely to interact with protein-coding genes." is not backed up by the data presented in this manuscript and need to be removed or rewritten. From the chapter starting on line 337 in the discussion, the authors use the term lncRNAs, while during the rest of the manuscript they use ncRNAs. The term ncRNA is correct as they have included small non-coding RNAs in their analysis. The heading also uses lncRNAs, which is not correct. The authors could exclude the small non-coding RNAs from their list of 929 non-coding RNAs in order to use the term lncRNAs throughout the paper.

Minor concerns:

The keyword "Non-coding RNAs prioritization" is confusing. Would this be a good search term for this paper?

Line 33: "early enhanced diagnostic techniques" poor wording. Is not sure what the authors mean by this statement. Further the whole sentence is poorly built up, and not well written. Please rewrite so that the sentence is clear.

Line 39: The word "cause" is not used correctly in the sentence. Maybe change to "in order"

Line 42: Insert "The" in front of "International Cancer Genome Consortium"

Line 47 needs a reference.

Line 50: Poor sentence, please rewrite "...our current understanding of genetic information's present functional consequences is highly primitive."

Line 54: The authors state that non-coding RNAs share sequence conservation as a trait with mRNA. Long non-coding RNAs are less conserved than mRNAs, and this should therefore be removed.

Line 181: DNase, change to DHS (also for the rest of the manuscript). Is written as DNase hypersensitive sites in line 188, as DNase in line 190, as DNase accessible sites in line 192

Line 186: The authors have written "2nd and 3rd set" differently, and should stick with one way of referring to these datasets.

Line 197: Remove "the" in front of "transcriptional regulation".

Line 238: "Expression quantitative trait loci" add abbreviation after.

Line 326: Should end with a "." and not a ";"

Reviewer #3 (Remarks to the Author):

In this article the authors analyze de novo somatic point mutations from 1855 breast cancer, using a newly developed tool. In this way they were able to identify 929 ncRNAs that are mutated in BC. Integrating these data with information taken by several databases the 929 ncRNAs contain marks for several regulatory features. In the end they provide a list of breast-cancer specific ncRNAs for better understanding genetic causes of breast cancer.

The study is well done, and all the analysis are performed following a stringent statistical analysis. The results are robust.

The article can be accepted for publication in its present form.

Rebuttal

We thank the reviewers for their constructive feedback. We made major changes to the manuscript in response to their comments that have improved the manuscript substantially. The main substantive issue raised by both reviewers was to analyse point mutations in different subtypes, compare ncRNAs with protein-coding genes and examine the expression of the ncRNAs in breast cancer. We have now substantially expanded the manuscript to address the above-mentioned points.

We generated 4 new Figures and 5 new Tables to address the reviewer's comments and provided a point-by-point response to the Reviewers' comments below, detailing changes that can already be made to the manuscript, along with details on how we will address specific issues raised by Reviewers.

Thank you for the opportunity to submit this preliminary plan of revision for our manuscript. We look forward to hearing from you in due course as to how to proceed.

Yours sincerely,

Hamid Alinejad-Rokny, Ph.D

(On behalf of the authorship team)

Reviewer #1

Reviewer #1 (Remarks to the Author):

The authors developed an R-based tool, dubbed as SomaGene, to identify 929 non-coding RNAs (ncRNAs) bearing significantly more somatic point mutations among >1800 patients diagnosed with breast carcinomas, compared to other types of cancer patients profiled by the ICGC. Through integrating multi-omics data including 1) chromatin states, histone modifications, TFBSs, and DNaseI hypersensitivity sites profiled by ENCODE, 2) actively transcribed promoters and enhancers identified by FANTOM5, 3) GWAS SNPs and eQTL polymorphisms, and 4) Hi-C assays for chromosome contact maps, the authors argued that these 929 ncRNAs have a great regulatory potential for protein-coding genes in breast cancers. In addition to breast tissue, the authors had deposited their analytical results for other five tissue types, including liver, blood, bone, lung, and skin, in their webtool to better share their findings with the ncRNA community. Despite their analysis pipelines showed little novelties and similar ideas can be found elsewhere, both standalone and online implementations displayed the results in a highly integrated manner which are beneficial to the ncRNA researchers interested in performing comprehensive analysis. My specific comments are listed below.

1) Breast cancer is a heterogeneous disease. Four well-established breast cancer subtypes (LumA, LumB, Her2+, and Basal-like) exhibited distinct molecular profiles and the underlying mechanisms driving the diseases are different between them. As ncRNAs are often expressed in a highly tissue- and context-specific fashion, I will expect each subtype

has its unique set of ncRNAs significantly enriched for somatic point mutations. Please include this information to your analysis throughout the manuscript.

Thank you for your valuable comments. We obtained PAM50 subtype annotation of 346 ICGC breast cancer samples from a publication by Nik-Zainal et al. In our list of significant ncRNAs, 782 hits had mutations in the above-mentioned samples. Most ncRNAs were mutated in samples of multiple subtypes, however, we observed that each subtype had a unique set of mutated ncRNAs that were not mutated in other subtypes figure below (Figure S2, Supplementary Table S2).

In addition, given the authors only studied somatic point mutations on the transcribed ncRNA transcripts rather than those across all non-coding DNA regions. Please examine if these 929 significant ncRNAs were expressed ubiquitously across all breast cancer patients or in certain subtypes only. If some of them were not expressed adequately, please reason how they exerted regulatory functions in breast cancer. Please also provide explanations why other mutations in non-coding DNA regions were excluded from your analysis.

We appreciate the reviewer’s comment. As we mentioned in the introduction, we only focused on those somatic point mutations that happened within the non-coding RNAs to identify significantly mutated ncRNAs in breast cancer.

To check if the significant ncRNAs were expressed ubiquitously across all breast cancer patients, we downloaded the expression dataset from Breast invasive carcinoma (BRCA) gene expression matrix from TANRIC (https://ibl.mdanderson.org/tanric/_design/basic/download.html). 504 ncRNAs of the 1030

significant ncRNAs were found in the TANRIC gene expression list. As the plot shows the 504 ncRNAs were expressed ubiquitously across all breast cancer patients (Figure below).

However, we had 106 ncRNAs that were differentially expressed between the breast cancer subtypes. As the Figure below (Figure S3) shows these ncRNAs have shown some breast cancer subtypes specificity. A list of these differentially expressed genes is provided in Supplementary table S3.

2) An expanded ncRNA catalogue is available. A recent work had expanded the catalogue of human ncRNAs with >3300 previously uncharacterized lncRNAs through profiling 300 human tissues and cell lines using PolyA+ and Total RNA-Seq (PMID: 34140680). Please include these novel lncRNAs to your analysis for the most updated results.

We appreciate the reviewer's comment. We have now added these new lncRNAs into our list and performed all the analyses. As a result, 76 new breast cancer associated lncRNAs were detected. We have now repeated all the enrichment analyses with the new set of candidate ncRNAs (1030 breast cancer associated ncRNAs).

Reviewer #1 (Minor comments):

1) While calling significance, please compare ncRNAs with the protein-coding genes for the five sections in Pages 4-8. This will help your readers understand how well these ncRNAs are associated with breast cancers over their protein-coding counterparts.

We appreciate the reviewer's comment. We have now repeated all the analyses on protein-coding genes as well. As the plot shows the enrichment of breast related regulatory features in the significant ncRNAs are more than protein-coding genes, indicating their key role as a regulatory element. We have now updated all figures (Figures 3-5) to incorporate these results.

2) For 226 ncRNAs supported by multiple Hi-C based assays, they should be separated from the significant set of 929 ncRNAs (or the 1st set) to see if they have better enrichment than other ncRNAs in the same set. In addition, please include the 2nd, 3rd, and last sets of ncRNAs in your Hi-C data analysis.

We appreciate the reviewer for their valuable suggestion. We have now performed a new analysis on Hi-C data to show if the interactions list of candidate ncRNAs have better enrichment than all significant ncRNAs. As supplementary figure S5 (below figure) shows the 226 ncRNAs (now 251 ncRNAs) supported by multiple Hi-C based assays have much higher enrichment of overlapping with regulatory features, GWAS and eQTL polymorphisms than other sets (e.g., significant set of ncRNAs, first, second and last set of ncRNAs). This supports our hypothesis that De novo somatic point mutations enriched in regulatory ncRNAs.

We have now incorporated these results into manuscript.

3) HMEC is a normal breast epithelial cell line. Please explain why MCF7 or other breast cancer cell lines profiled by ENCODE were not considered.

We thank the reviewer comment. We would love to replicate our Hi-C analysis on a breast cancer cell line Hi-C data and compare them with a healthy cell line. Unfortunately, we were not able to find a high resolution Hi-C data (like HMEC cell line) for breast cancer cell lines. We, however, look forward to performing such the analysis in the future.

4) How many GWAS SNPs and eQTL polymorphisms were overlapped with the somatic point mutations? If a significant overlap is observed, please explain if the results in Pages 7-8 are essentially the product of circularity biases. Conversely, please discuss if both germline and somatic mutations are required to develop breast cancers.

Thank you for your comment. We analysed the overlap of somatic point mutations with ~470'000 eQTL polymorphisms (associated with breast mammary tissue) and ~1000 GWAS SNPs (with mapped trait related to breast cancer). Among ~4'000'000 somatic point mutations, only ~500 and 4 mutations overlapped with eQTL polymorphisms and GWAS SNPs respectively.

We have also discussed the importance of both germline and somatic mutations in breast cancer development in discussion section of the paper.

Reviewer #2

Reviewer #2 (Remarks to the Author):

The authors have investigated the role of somatic mutation found within non-coding RNA genomic regions in breast cancer, and further investigate the association of mutated non-coding RNAs with several features, including annotated promoters/enhancers, histone modification, CTCF binding sites, DNase hypersensitive sites, GWAS SNPs, eQTL, and Hi-C. The manuscript includes several well performed in silico analysis and the method section is satisfactory written. The language is of good quality, but there are multiple sentences with poor quality that needs to be rewritten. Some of the figures needs also to be better presented. The peer reviewers major concern is that for the association analysis, most of these features included are associated with functional DNA, and less with functional RNA. These analyses are extremely interesting for identifying new DNA regulatory elements, but the association towards non-coding RNA transcripts is lacking. The manuscript therefor needs to be revised in order to better state how these analyses are relevant for non-coding RNAs. Some new analysis regarding the RNA expression needs to be performed, and the manuscript must in its introduction give an introduction towards the subtypes of enhancer and promoters non-coding RNAs. All of my concerns are more thoroughly addressed below.

1) In abstract line 10, the authors use of the word “explicitly”. This word can make the reader think that the mutations are exclusively in breast cancer, which the peer reviewer find them not to be based on the presented data. The mutations were significantly when compared to an average of multiple other cancers, but this does not exclude that they could also have been significantly enriched in other specific type of cancer. The number of patients with a specific mutation found in a non-coding region, should be presented in a plot where the other cancers are separated into their individual subtypes, in order to check if the mutations were breast cancer specific, or were also specific to other cancer types. In line 71, the authors use the term specifically, which I feel is not thoroughly backed up based on how the comparison was done (compared against a pan-cancer average).

We appreciate the reviewer’s comment. We then checked if the mutations in the candidate set of ncRNAs are breast cancer specific or these ncRNAs are also significantly mutated in other cancers. To do this, we first identified the candidate set of ncRNAs in 17 other cancer types.

We then calculated how many of the 1030 significant ncRNAs in breast cancer are significant in other cancers (Table 1a). As the table shows, at least 427 of 1030 ncRNAs are explicitly mutated in breast cancer samples, only.

Also, we sorted the significantly mutated ncRNAs in each cancer based on their mutational P-value and took 1030 top ncRNAs to see how many of them are common with the 1030 BC-specific ncRNAs. This analysis will reveal if the 1030 BC-associated ncRNAs are more BC specific or not. As Table 2b shows at least 916 (%80) of the ncRNAs in our BC-associated list were not in the top 1030 significantly mutated ncRNAs of other cancers, indicating the 1030 ncRNAs identified in our study are more relevant to breast cancer than other cancers (e.g., more breast cancer specific).

We have now incorporated these results into our paper.

Table1. Number of significantly mutated ncRNAs in breast cancer and other cancers. **a)** Each number in the table shows that how many of the 1030 significant ncRNAs in breast cancer are also significant in other cancer. **b)** Each number in the table shows that how many of top 1030 candidate ncRNAs in other cancers are in the BC-associated ncRNAs.

bladder	blood	bone	brain	breast	cervix	colorectal	esophagus	kidney	liver	lung	ovary	pancreas	prostate	skin	stomach	uterus
Table 1a. Number of significant ncRNAs in each cancer that are common with BC-associated ncRNAs																
3	94	0	0	1030	3	7	277	1	176	79	263	230	91	603	3	16
Table 1b. Top 1030 significant ncRNAs in each cancer that are common with BC-associated ncRNAs																
6	58	12	0	1030	5	1	6	1	17	30	116	97	87	8	3	20

2) Figure 1 and the figure legend is not corresponding well. The figure should have included A,B,C, and D, in order to point to where in the figure the legend is referring to. The same concern is also for figure 2, where the figure is not easy to interpret from the figure legend, and does not give the reader any extra information regarding the results or better understanding of how the data was generated. The authors should carefully go through the figures and do modifications, as well as modify the figure legends so that they explain the figures and their data better. In figure 4, please explain the S (score threshold). Figure 3, 4, and 5 have the same figure heading in the figure legends.

We appreciate the reviewer comment. We have now modified figures and provided more information in the figure legends.

3) The chapter “Enrichment of regulatory features in the BC-associated ncRNAs” might not have a good heading, as it suggests that multiple regulatory features will be examined in this chapter, while only enhancers/promoter regions are examined. It is also not clear from the first lines in the introduction to this chapter what type of features they are examining, and the authors should state why they assign these types of regions as regions important for non-coding RNAs with regulatory functions. Also the 2nd and 3rd sets of ncRNAs should be better described when first introduced in this chapter. It would have been beneficial to include a figure showing the overlap between the numbers of non-coding RNAs identified by ENCODE and FANTOM (not only as a supplementary table), to see their overlap.

We appreciate the reviewer’s comment. We have now added a new paragraph before the result section to better describe 2nd, 3rd, and last sets of ncRNAs. We also added a new table (supplementary table S7) to show the overlap between candidate ncRNAs with ENCODE and FANTOM5 features. Notably, there were 317 ncRNAs that overlapped with both ENCODE and FANTOM5 enhancer marks and 257 ncRNAs that overlapped with both ENCODE and FANTOM5 promoters.

4) For the chapter “Histone modifications H3K27ac and H3K4me1, CTCF binding sites, and DNase hypersensitive sites are significantly enriched for BC-associated ncRNAs” the authors should give a more specific description regarding the markers and how these markers are relevant for non-coding RNAs. Here, it would be interesting to see the overlap with these features across each other and with the enhancer/promoter analysis performed in the chapter above, as this could point towards specific non-coding RNAs being highly associated with DNA regulatory regions. Personally, I would have been focusing more on the non-coding RNAs not being enriched by these features and the enhancers/promoters, as the features have a stronger association towards regulatory DNA regions than functional RNAs. Better description on how these features have previously been by the literature used to identify functional RNAs, would help the authors convince the readers about the relevance for conduction these types of correlation analysis.

We appreciate the reviewer’s comment. We have now provided a description of the importance of these markers. As the reviewer mentioned, the overlap between candidate ncRNAs and the regulatory markers was provided in supplementary tables. However. We also provided a new figure (supplementary figure S4) to show the overlap between candidate ncRNAs and the regulatory markers.

5) The chapter “BC-associated GWAS SNPs are significantly enriched in the candidate non-coding RNAs” is more a confirmation of the results from the section “Background model to identify significant non-coding RNAs in breast cancer” and should be written together as a single chapter in the beginning of the manuscript, or come as an independent chapter right after the first chapter. Similarly, the chapter “BC-associated non-coding RNAs have a significantly higher fraction of eQTL polymorphisms” should be moved up as this also gives evidence towards these non-coding RNAs being important in disease.

Thank you for your comment. We have now re-positioned these sections. As the reviewer mentioned, in the supplementary tables, we indicated those ncRNAs that overlap with these features. However, we also added a new figure (supplementary figure S4) to show the overlap between candidate ncRNAs and all the features.

6) The chapter “Chromosome conformation capture data shows a potential regulatory role for BC-associated non-coding RNAs” includes an analysis where the DNA is in focus, and where it is a lack in understanding how this analysis gives evidence for a functional RNA. Also for this analysis, it would be beneficial to present the overlap with the previous analysis in a figure. The proposed inclusion of expression analysis of the non-coding RNAs, could be supplemented with correlation analysis of expression values between the non-coding RNAs with their chromosomal interacting protein coding genes.

We thank the reviewer for their comments. The reviewer is correct, and Hi-C is not the best technique to investigate RNA-DNA interactions. To identify interactions between ncRNAs and genes, we need data from RNA-DNA technologies (e.g., RADICAL-seq and GRID-seq). However, such data is not available for a breast related cell line.

In our Hi-C analysis, we were interested to see if the genomic loci that overlaps/encompass the ncRNAs has an enhancer-like activity. We have now revised the text to clarify this confusion.

We have also added a new figure (Supplementary figure S4) that shows the overlapping of 251 candidate ncRNAs with genomic and epigenetic features used in this study.

We also checked if the 251 ncRNAs (with Hi-C support in the both Hi-C libraries) are expressed across TANRIC breast cancer samples at a different level compared to the rest of ncRNAs. To do this, for each sample, the overlap of 251 ncRNAs with the top 500 ncRNAs with highest expression was calculated. The significance of observed overlaps per each sample were assessed using one sided binomial test with the number of trials being 500 and the probability of success being the percentage of ncRNAs in TANRIC dataset that match the ncRNAs in the 251 ncRNAs. All per sample p-values were adjusted for multiple testing using Benjamini-Hochberg method. However, as the below figure shows the adjusted p-values is not significant at the significantly threshold of 0.05. (Group 1 refers to 251 significant ncRNAs and Group 2 refers to all non-significant ncRNAs).

We have now incorporated these results into the paper.

7) The statement “Our analysis demonstrated that highly mutated lncRNAs have more cell type-specific functions and are more likely to interact with protein-coding genes.” is not backed up by the data presented in this manuscript and need to be removed or rewritten. From the chapter starting on line 337 in the discussion, the authors use the term lncRNAs, while during the rest of the manuscript they use ncRNAs. The term ncRNA is correct as they have included small non-coding RNAs in their analysis. The heading also uses lncRNAs,

which is not correct. The authors could exclude the small non-coding RNAs from their list of 929 non-coding RNAs in order to use the term lncRNAs throughout the paper.

Thank you for your comment. We have now re-written the sentence. We also replaced lncRNAs with ncRNA where applicable.

Reviewer #2 (Minor comments):

1) The keyword "Non-coding RNAs prioritization" is confusing. Would this be a good search term for this paper?

Thank you for your comment. We have now updated the keyword lists.

2) Line 33: "early enhanced diagnostic techniques" poor wording. Is not sure what the authors mean by this statement. Further the whole sentence is poorly built up, and not well written. Please rewrite so that the sentence is clear.

Fixed.

3) Line 39: The word "cause" is not used correctly in the sentence. Maybe change to "in order".

Fixed.

4) Line 42: Insert "The" in front of "International Cancer Genome Consortium".

Fixed.

5) Line 47 needs a reference.

Fixed.

6) Line 50: Poor sentence, please rewrite "...our current understanding of genetic information's present functional consequences is highly primitive."

Fixed.

7) Line 54: The authors state that non-coding RNAs share sequence conservation as a trait with mRNA. Long non-coding RNAs are less conserved than mRNAs, and this should therefore be removed.

Fixed.

8) Line 181: DNase, change to DHS (also for the rest of the manuscript). Is written as DNase hypersensitive sites in line 188, as DNase in line 190, as DNase accessible sites in line 192.

Fixed.

9) Line 186: The authors have written "2nd and 3rd set" differently, and should stick with one way of referring to these datasets.

Fixed.

10) Line 197: Remove "the" in front of "transcriptional regulation".

Fixed.

11) Line 238: "Expression quantitative trait loci" add abbreviation after.

Fixed.

12) Line 326: Should end with a "." and not a ";

Fixed.

Reviewer #3

Reviewer #3 (Remarks to the Author):

In this article the authors analyze de novo somatic point mutations from 1855 breast cancer, using a newly developed tools. In this way they were able to identify 929 ncRNAs that are mutated in BC. Integrating these data with information taken by several db the 929 ncRNAs contain marks for several regulatory features. In the end they provide a list of breast-cancer specific ncRNAs for better understanding genetic causes of breast cancer.

The study is well done, and all the analysis are performed following a stringent statistical analysis.

The results are robust.

The article can be accepted for publication in its present form.

We appreciate the reviewer for their positive comment.

REVIEWERS' COMMENTS:

Reviewer #1 (Remarks to the Author):

Each point raised in my previous review has been adequately addressed, and the quality of manuscript has been significantly improved. I am satisfied with the authors' responses to my comments as well as another reviewer's.

Reviewer #2 (Remarks to the Author):

The authors have made satisfactory corrections to their manuscript and have adequately addressed all the concerns raised in the peer reviewing process. The manuscript is therefore found to be suitable for publication.

The peer reviewer has two minor concerns that should be corrected before publication:

1) The PAM50 subtypes include five subtypes: Luminal A, luminal B, HER2-enriched, basal-like, and normal-like. In the text the authors have only referred to the four pathological subtypes, and in the figure, they have included the five PAM50 subtypes. Here they have named the "normal-like" subtype for "normal". It is important to not define it as only "normal" as this can indicate that it is a normal sample and not a cancerous sample. Please correct this minor mistake and refer to the two original papers when describing the PAM50 subtyping in the text.

2) The manuscript does still contain multiple minor language issues, especially in the newly added text. We do encourage that the manuscript is thoroughly read through before final publication or submitted for professional language editing.